# Comparison of greenhouse gas emissions associated with the construction of timber, concrete, and steel check dams in Akita, Japan: An input-output analysis

Tomohumi Huzita[1][*], Ryu Noda[2]

1 Nagano University, Ueda, Nagano, Japan, 2 Institute of Wood Technology, Akita Prefectural University, Noshiro, Akita, Japan

☯ These authors contributed equally to this work.
* huzita@nagano.ac.jp

## Abstract

To mitigate global warming, replacing concrete and steel with timber as the primary construction material for construction projects, such as check dams, is being promoted in Japan and other countries. Timber check dams have more limited installation sites than concrete or steel dams because of installation conditions such as locations less susceptible to debris flows and locations where there is constant running water. However, even when the installation conditions are met, engineers and contractors are reluctant to select timber as a construction material because of its high construction cost. In this study, an input-output table was used to compare the greenhouse gas (GHG) emissions associated with the construction of a timber check dam at the design stage with those associated with the construction of concrete and steel check dams to quantitatively evaluate the added value of timber utilization (in addition to its construction cost). The results revealed that replacing concrete and steel check dams with timber check dams could reduce GHG emissions by 61% and 34%, respectively. This study demonstrated the possibility of evaluating the GHG emissions associated with a construction project at the design stage. Moreover, it highlights the importance of considering the GHG emissions associated with construction materials when selecting the most appropriate materials for public works projects.

## Introduction

The Intergovernmental Panel on Climate Change's (IPCC) Special Report on Global Warming of 1.5°C, released in October 2018, stated that to limit global warming to 1.5°C above pre-industrial levels, net greenhouse gas (GHG) emissions must reach zero around 2050. To limit global warming to 2.0°C, net GHG emissions should reach zero by approximately 2070 [1]. However, according to the Synthesis Report of the IPCC's Sixth Assessment Report, which was released in March 2023, the climate change measures implemented by individual countries before the 26[th] Conference of the Parties to the United Nations Framework Convention on

**Data Availability Statement:** The data underlying the results presented in the study are available from Statistics Bureau of Japan, https://www.e-stat.go.jp/en. The data underlying the results

presented in the study are available from National Institute for Environmental Studies, https://www.cger.nies.go.jp/publications/report/d031/eng/index_e.htm.

**Funding:** Initials of the author: TH Grant number: KI0242055 The full name of each funder: Yanmar Environmental Sustainability Support Association URL: https://www.yanmar.com/jp/shigenjunkan/index.html The funders had no role in study design, data collection and analysis, decision to publish, or preparation of the manuscript.

**Competing interests:** The authors have declared that no competing interests exist.

Climate Change contribute to the rise of global temperature beyond the 1.5°C target set by the Paris Agreement [2]. In this context, the 49th Summit of the Group of Seven (G7) industrialized nations, held in May 2023 in Hiroshima, Japan, emphasized the urgency of reducing GHG emissions by 43% by 2030 and 60% by 2035 to achieve the 1.5°C target [3]. The goal of achieving net-zero emissions by 2050 was set during the said summit, and the G7 nations requested other countries that are far from realizing the 1.5°C target to increase their climate change mitigation efforts. Considering the aggravating impacts of climate change, reducing GHG emissions is an important issue for Japan, a member of the G7 nations.

Concrete and steel are the commonly used materials in construction and civil engineering projects. Cement, the raw material for concrete and iron, emits a large amount of GHG during production. In particular, the production of cement, which is the raw material for concrete, contributes to approximately 5% of the global anthropogenic GHG emissions [4]. Of course, GHG emissions from the production of cement and iron have been reported quantitatively in recent years, and it has also been reported that emissions are reduced when material substitution, material reduction, and recycling are considered [5–7]. However, GHG emissions associated with cement production may increase in the future, owing to the rising demand for cement in developing countries [8].

Furthermore, the GHG emissions associated with steel products have also been reported to increase owing to the expansion of steel production in emerging countries, despite technological improvements in some regions [9]. Therefore, reducing their GHG emissions would be challenging for the cement and steel industries [10, 11]. To mitigate global warming, substituting concrete and steel products with other materials with a lower environmental impact should be considered. For example, a report comparing the environmental impact assessment of structural timber with concrete and steel using life cycle assessment reported that timber has a lower impact on global warming than concrete and steel with equivalent functions [12–14].

It is recommended that timber be used from sustainable forests. Indeed, there is concern about the effects of deforestation on wildlife and climate change in Southeast Asia and Africa [15–17]. However, 40% of Japan's forests are planted forests, many of which were planted after World War II with the expectation of timber use [18]. Therefore, many forests are reaching the end of their harvesting season, and it is said that forests need to be renewed by using them as timber and reforesting them [18]. The "Act for Promotion of Use of Wood in Buildings to Contribute to the Realization of a Decarbonized Society" has been enacted in Japan to promote the use of wood, which has a lower environmental impact, in constructing buildings and public facilities as an alternative to conventional concrete and steel materials. Reduction in GHG emissions due to replacing concrete with wood has been reported [19, 20]. In Japan, timber is mostly used for constructing buildings; however, due to the declining population, the number of architectural projects in Japan has recently decreased, resulting in lower demand for timber in building construction [18]. Hence, the potential application of timber in civil engineering is being explored [18, 21, 22].

The use of timber in civil engineering projects has been quantitatively evaluated in terms of its environmental and economic impacts. O'Born [23] and Son et al. [24] reported that using timber for making bridges was associated with lower GHG emissions than concrete. Johanne et al. [25] conducted an environmental life-cycle assessment of Norway's concrete, steel, and timber bridges and reported that timber contributed less to global warming than concrete and steel. In contrast, Hassan et al. [26], Iwase et al. [27], and Noda et al. [28] found that although the replacement of concrete with timber as a construction material reduces GHG emissions, it increases construction costs; thus, timber substitution should be quantitatively evaluated in terms of a trade-off between GHG emissions and construction cost. Moreover, Shafayet et al. [29] reported that mass timber construction is more expensive than concrete construction due

to the high cost of timber processing and staffing. Scouse et al. [30] and Huzita et al. [31–33] evaluated the impact of timber substitution on regional economies. They quantitatively assessed the ripple effects of timber utilization on regional industries and job creation. Their findings indicate that although using timber in civil engineering increases construction costs, it is expected to reduce GHG emissions and stimulate regional economies. Even though it has been reported that reducing GHG emissions may increase the economic burden and reduce economic growth [34], the use of timber in the civil engineering sector is favorable because of its low environmental and economic impact.

Timber is not often selected as a construction material for civil engineering projects in Japan because it is more expensive than other construction materials. Usually, the GHG emissions associated with a construction project are assessed by determining the emissions generated during the production and combustion of fuels and the manufacture of construction materials used in the project (process flow diagram). However, this evaluation approach makes it difficult for structural designers to assess the GHG emissions associated with construction projects at the design stage. If GHG emissions can be evaluated as an added value of construction material at the design stage of a construction project, a quantitative comparison between the GHG emissions associated with timber and other types of construction materials can be made possible, which allows for comparisons that are not solely based on the construction costs.

The "New Ministry of Agriculture, Forestry and Fisheries Wood Utilization Promotion Plan" [35] advocates for the utilization of thinned wood; moreover, the "Guidelines for Design of Wooden Structures for Forest Engineering" [36] and "Tentative Construction Procedure for Wooden Structures for Forest Engineering" [37] have been established to promote the use of timber in the field of forest engineering.

In this study, we aimed to determine the GHG emissions associated with constructing a timber check dam, a forest engineering structure, at the design stage. We compared these emissions with those associated with concrete and steel check dams to evaluate whether timber is sustainable during the design phase. A timber check dam was used as the research subject because it contains more timber per unit area, and timber does not easily decay when submerged in water. Specifically, an input-output table was used to evaluate GHG emissions based on the values in the design documents for timber, concrete, and steel check dams. In this study, we aimed to compare the GHG emissions from various materials used in the construction industry to select the most suited material for public construction projects based on its environmental sustainability. The results obtained in this study indicate that we were able to achieve this objective successfully.

## Materials and methods

### Region

The area of interest in this study was Akita Prefecture, located in northern Japan. Akita Prefecture is a major producer of Japanese cedar, commonly used as a construction material in Japan. Cedar production accounts for approximately 90% of the log material production in Akita Prefecture [38]. Akita has the largest Japanese cedar plantations among the Japanese prefectures. In addition, the guidelines for promoting wood use are based on the Akita Prefecture Wood Use Promotion Ordinance, which promotes using timber in forest engineering projects in Akita [39].

### Check dams

A check dam is a forest engineering structure installed to prevent stream erosion; it stabilizes the stream bed and prevents unstable sediments from flowing. Approximately 80 check dams

**Table 1. Scale of check dams.**

| Check dam | Length | Height | Crown width | Volume |
|---|---|---|---|---|
| | (m) | (m) | (m) | (m$^3$) |
| Timber | 20.7 | 3.00 | 1.80 | 156.6 |
| Concrete | 21.0 | 4.00 | 1.50 | 150.5 |
| Steel | 22.0 | 4.00 | 1.50 | 154.7 |

are constructed annually in Akita Prefecture [38]; check dams are important structures in Akita Prefecture, which has several mountainous areas. This study evaluated the GHG emissions associated with average-sized timber, concrete, and steel check dams. Regardless of the construction material used, the service life of check dams is 50 years. Table 1 presents the scale of the target check dams.

During this study, the GHG emissions from the time of production of construction materials to the actual construction of the check dams were evaluated by examining the design documents. It was assumed that the check dams would not be maintained, disposed of, or recycled after installation. Maintenance was not considered a completely predictive evaluation that may have produced results different from the real situation, provided that external factors, such as mountain disasters and the location of dam installation, significantly affect the integrity of check dams; moreover, maintenance was not described in the design documents. Disposal and recycling were also not considered because check dams are seldom removed even after their service life has ended. Furthermore, GHG emissions have been evaluated by only focusing on the construction phase of a civil engineering project [40–43]; hence, this study did not consider the maintenance, disposal, and recycling of check dams.

## Input-output table

The input-output table containing the transactions valued at the producers' prices in 2015 (107 sectors) [44] was used as the reference in this study. However, the industrial sector was subdivided into major sectors to comprehensively calculate the GHG emissions associated with the major sectors, with a particularly high contribution to each check dam's construction (Table 2).

## Input-output analysis

**Scope of evaluation.** The scope of the evaluation of GHG emissions using the input-output table was assumed to have direct and indirect effects. Direct effects pertain to the GHG emissions generated in Akita Prefecture due to the final demand increase. Direct effects represent the GHG emissions associated with the fuel consumed by heavy equipment at construction sites and the manufacture of construction materials used in construction projects. Indirect effects pertain to GHG emissions induced by the direct effects of each industry. Indirect effects represent the GHG emissions associated with producing raw materials required for manufacturing the construction materials used in a construction project. To illustrate, indirect effects pertain to the GHG emissions associated with cement production and aggregates, raw materials required to manufacture ready-mixed concrete. The production values effectuated by the direct and indirect effects of the check dams were evaluated to evaluate the GHG emissions associated with check dams. The production value indicates the amount of output effectuated in each industry to cover the final demand generated in each industry. The target GHGs in this study were carbon dioxide ($CO_2$), methane ($CH_4$), dinitrogen monoxide ($N_2O$), hydrofluorocarbons, organofluorine compounds, sulfur hexafluoride, and nitrogen trifluoride.

**Table 2. Sectoral divisions of major industrial sectors.**

| Categories of major sectors | New sectors |
|---|---|
| **Forestry** | Silviculture |
| | Logs |
| | Special forest products (including hunting) |
| **Lumber and wood products** | Timber |
| | Plywood and glued laminated timber |
| | Wooden chips |
| | Miscellaneous wooden products |
| **Cement and cement products** | Cement |
| | Ready-mixed concrete |
| | Cement products |
| **Steel products** | Hot-rolled steel |
| | Steel pipes and tubes |
| | Cold-finished steel |
| | Coated steel |
| **Miscellaneous metal products** | Gas and oil appliances and heating and cooking apparatus |
| | Bolts, nuts, rivets, and springs |
| | Metal containers, fabricated plates, and sheet metal |
| | Plumbing accessories and powder metallurgy products and tools |
| | Miscellaneous metal products |
| **Goods rental and leasing services** | Goods rental and leasing |
| | Car rental and leasing |

**Final demand increase.** The final demand increase was defined as the cost of materials, machinery, and equipment stated in the design document for the construction of the check dams, and each cost was categorized under its corresponding industry sector. The check dam was designed in 2024; however, the most recent input-output table was published in 2020 and represents the 2015 industrial structure. To unify the years when the check dams were designed, and the input-output table was published, the prices of construction materials in 2024 were converted to those in 2015 using the Construction Material Price Index (Sendai) [45].

The design document contains the prices set by purchasers, including the commercial and transportation margins. As the input-output table uses the prices set by producers, the purchasers' prices were converted into producers' prices by removing the commercial and transportation margins. Therefore, the commercial margin (wholesale and retail) and domestic freight rates for each sector in the total domestic demand in the input-output table (basic sector) [44] were divided by the purchasers' prices to obtain the commercial margin and domestic freight rates. The producers' prices for each expense category were calculated by multiplying the commercial margin and domestic freight rates obtained for the material, machinery, and equipment costs. The removed margins were categorized under their corresponding commerce and transportation sectors. Table 3 shows the final demand increase in producers' prices in each industry sector (materials, machinery, and equipment costs). The materials used for each dam and the corresponding industrial sectors are shown in S1–S3 Tables.

## GHG emissions

This study includes raw materials, transportation, material production, and construction of check dams. The system boundaries for constructing timber, concrete, and steel check dams are shown in Fig 1.

**Table 3. Final demand increases (in Japanese yen) in each industrial sector.**

| Industrial sector | Check dam | | |
|---|---|---|---|
| | **Timber** | **Concrete** | **Steel** |
| Crop cultivation | 103,184 | 41,322 | 22,449 |
| Miscellaneous mining industry | 213,998 | 0 | 716,002 |
| Textile products | 48,935 | 445 | 15,868 |
| Timber | 6,247,959 | 626,288 | 2,917 |
| Petroleum refinery products | 78,986 | 46,711 | 96,340 |
| Plastic products | 310,961 | 313,636 | 310,961 |
| Ready-mixed concrete | 0 | 1,963,460 | 0 |
| Miscellaneous ceramic, stone, and clay products | 0 | 6 | 0 |
| Hot-rolled steel | 29,249 | 0 | 287,372 |
| Coated steel | 0 | 0 | 3,277,657 |
| Non-ferrous metal products | 0 | 35,919 | 35,919 |
| Bolts, nuts, rivets, and springs | 2,285,435 | 0 | 0 |
| Commerce | 1,726,981 | 787,917 | 615,812 |
| Railway transport | 1,160 | 198 | 736 |
| Road transport (except self-transport) | 724,274 | 187,857 | 163,453 |
| Water transport | 54,436 | 5,249 | 50,384 |
| Air transport | 103 | 31 | 21 |
| Consigned freight forwarding | 46,325 | 5,035 | 12,581 |
| Storage facility service | 45,728 | 6,224 | 29,987 |
| Goods rental and leasing services | 737,909 | 131,248 | 269,877 |
| Total | 12,655,623 | 4,151,546 | 5,908,336 |

The procedure for evaluating the GHG emissions associated with the construction of the check dams is shown in Fig 2.

The input coefficient matrix ($A$) has been expressed in Eq (1) and was calculated using $a_{ij} = \frac{x_{ij}}{X_j}$:

$$A = \begin{bmatrix} a_{11} & a_{12} & \cdots & a_{1n} \\ a_{21} & a_{22} & \cdots & a_{2n} \\ \vdots & \vdots & \ddots & \vdots \\ a_{n1} & a_{n2} & \cdots & a_{nn} \end{bmatrix} \tag{1}$$

where $i$ is the row sector number, $j$ is the column sector number, $a_{ij}$ is the input coefficient, $x_{ij}$ is the amount of input of sector $j$ from sector $i$, and $X_j$ is the country's output of sector $j$.

The inverse matrix coefficient ($B$) was obtained as $B = [I - (I - M)A]^{-1}$, where $I$ is the unit matrix, and $M$ is the diagonal matrix of import coefficients. An open-economy-type inverse matrix reflects that some spillover effects flow out of the country as imports from abroad were used. The import coefficient was obtained by dividing each sector's imports by the total domestic demand. However, because the cedar used for the timber check dam was produced entirely in Akita Prefecture, and no imported timber was used, the import coefficients for the timber and log sectors of the construction of the timber check dam were set at 0.

The GHG emissions associated with the check dams were calculated by multiplying the production value induced in each industrial sector by the GHG emission intensity, which was calculated using the unit direct GHG emissions reported in the Embodied Energy and Emission

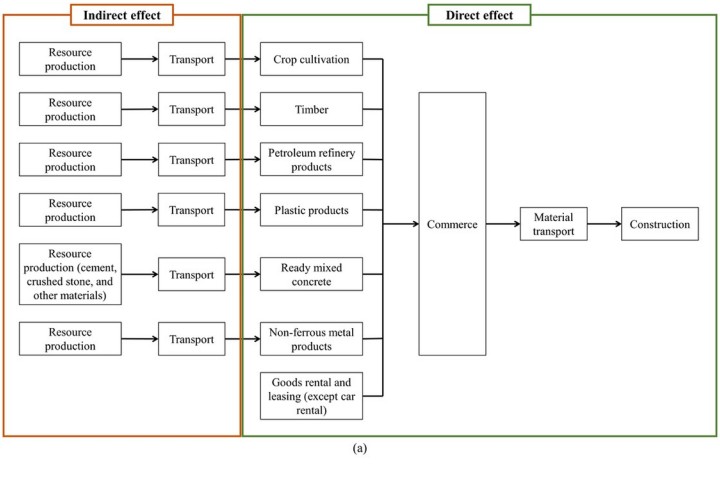

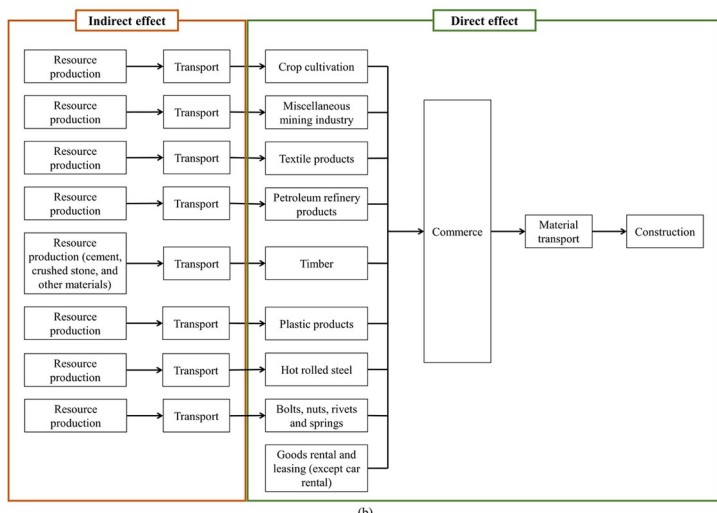

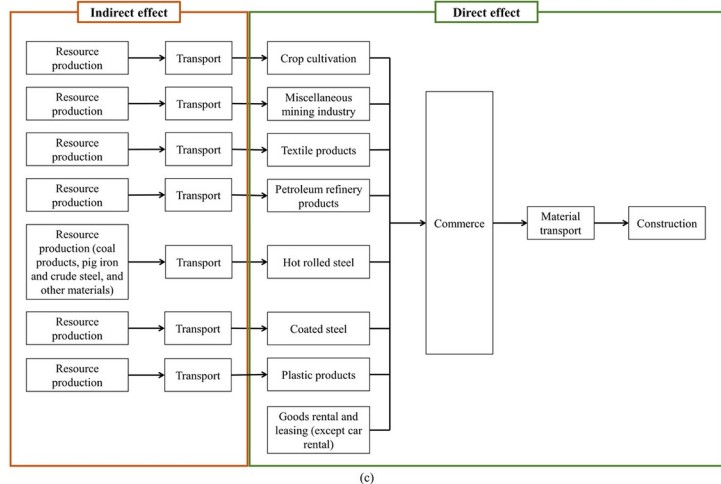

**Fig 1.** System boundaries of check dam construction: (a) timber check dam, (b) concrete check dam, (c) steel check dam.

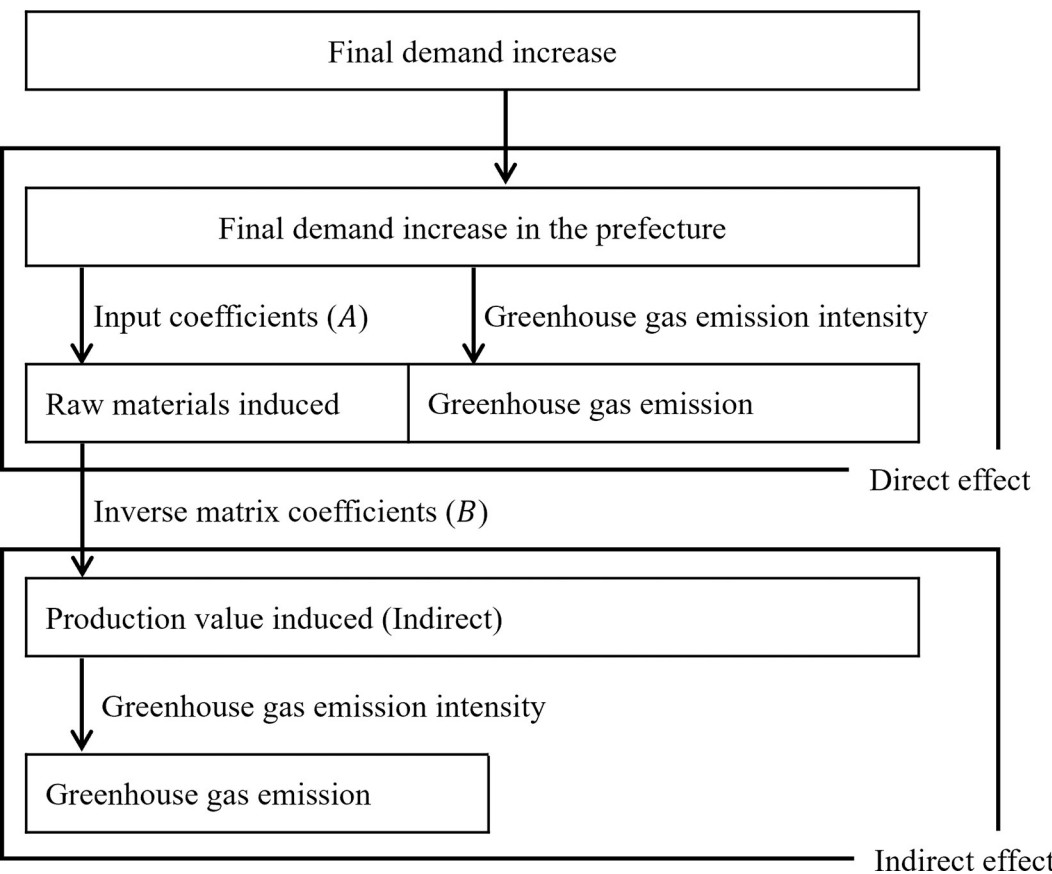

**Fig 2. Procedure for greenhouse gas emissions assessment.**

Intensity Data (3EID), following the calculation guidelines developed by the IPCC and agreed upon by the Conference of the Parties [46, 47]. The 390 industry sectors included in the 3EID were merged into 122 sectors according to the sector classification in the split input-output table.

## Results and discussion

### GHG emission

Table 4 shows the GHG emissions associated with constructing timber, steel, and concrete check dams. The table includes values of the direct and indirect effects, total effects, and unit

**Table 4. Greenhouse gas emissions (measured in kg-CO$_2$eq) are associated with constructing timber, concrete, and steel check dams.**

| Check dam | Timber | Concrete | Steel |
|---|---|---|---|
| Direct effect (D) | 4,579 | 2,120 | 3,680 |
| Indirect effect (I) | 19,549 | 46,902 | 32,121 |
| Total (D+I) | 24,128 | 49,022 | 35,801 |
| Unit emission of direct and indirect effects [a] (kg-CO$_2$eq/thousands-JPY) | 1.91 | 11.81 | 6.06 |

[a]The greenhouse gas emissions effectuated by the direct and indirect effects of the check dams were divided by the final demand increase.

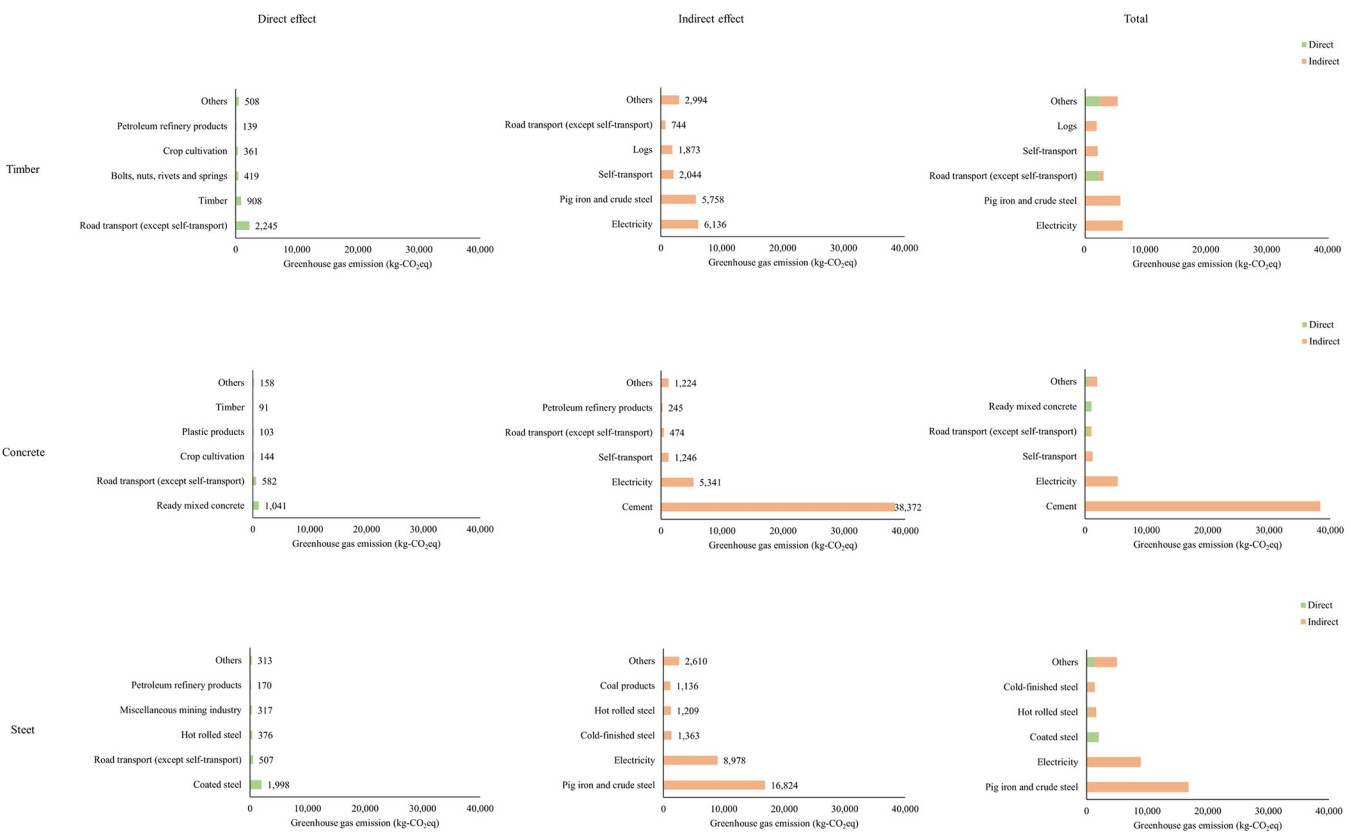

**Fig 3. Top five sectors contributions to the direct and indirect effects of timber, concrete, and steel check dam construction.**

emission effectuated by the direct and indirect effects of the timber, concrete, and steel check dams. Regarding GHG emissions, the timber check dam was more advantageous than the concrete and steel check dams. The total GHG emissions associated with the concrete and steel check dams were 2.03- and 1.48-times, respectively, higher than those associated with the timber check dam. The timber check dam was also superior in terms of GHG emissions per unit for the final demand increase. The results suggest that the added value of timber check dams can be quantified because the construction of the timber check dam was associated with fewer GHG emissions than the other check dams.

Fig 3 shows the top five sectors contributing to timber, concrete, and steel check dam construction's direct and indirect effects. Please refer to S4–S6 Tables for further details.

Among the three types of check dams, the GHG emissions effectuated by the direct effects of timber check dams were the highest. In particular, the road transport (except self-transport) sector accounted for 49% of the direct effects of timber check dam construction. Regarding direct effects, the high GHG emissions associated with the other dams could be attributed to the production of the main construction materials (coated steel and ready-mixed concrete). In contrast, the high GHG emissions associated with the timber check dam could be attributed to transporting timber from the timber processing plant to the construction site. Therefore, although small amounts of GHGs are emitted during timber production, large amounts of GHGs are emitted during its transportation. As timber is lighter than concrete and steel, emissions related to transportation are expected to be reduced; however, the cost of the transportation of timber is higher than that of steel and concrete (Table 3). Thus, it is possible that during transportation, the volume of timber (including spaces and voids) was larger than its

actual volume, resulting in higher GHG emissions. For example, it has been reported in construction that the transportation distance of timber affects GHG emissions [48]. It has also been reported that GHG emissions are high in log production, the raw material for lumber, during yarding and stacking [49]. The results of these previous studies are consistent with the high GHG emissions of timber in the transportation process obtained in this study; this suggests that shortening transportation distances, such as by using regional timber, is also important to reduce GHG emissions from timber structures.

The GHG emissions effectuated by the indirect effects of the concrete check dam were the highest, followed by those of the steel and timber check dams. A high proportion of the GHG emissions associated with all check dams was attributed to the electricity sector, indicating high electricity consumption while producing raw materials. However, approximately 23% and 45% of the indirect emissions from the electricity consumption of the timber check dam were attributed to the production of timber and metal fittings, respectively. As the production of metal fittings contributed to the high GHG emissions of the timber check dam, reducing the amount of metal fittings would be effective at further reducing the GHG emissions associated with timber check dams.

The emissions from the cement sector accounted for 78% of the total GHG emissions associated with concrete check dams, indicating that GHG emissions associated with cement production were particularly high when fresh concrete was used. However, this study's input-output table provided an overview of the 2015 industrial structure. When the concrete check dam was designed in 2024, emissions from cement production may have been reduced owing to improved energy efficiency in cement production and the development of low-carbon cement [50, 51]. For example, it is reported that a large amount of GHGs are emitted in the production of clinker, a raw material for cement, and that reducing clinker content is important for reducing GHG emissions [52]. Reduced GHG emissions have been reported using alternative materials (silica fume and fly ash) as part of cement [53]. Watari et al. [54] showed that with demand- and supply-side mitigation efforts, $CO_2$ emissions from Japan's entire cement and concrete cycle may be significantly reduced by 2050. Assuming that the $CO_2$ emissions associated with cement production were 1.000 in 2015, the $CO_2$ emissions associated with cement production in 2024, based on the scenarios analyzed in a previous study [54], were estimated to be 0.829 for the baseline, 0.763 for the supply-side strategies, and 0.635 for the supply and demand side strategies. In this study, the GHG emissions associated with concrete check dam construction were evaluated, assuming that $CO_2$ emissions associated with cement production have decreased since 2015; however, no changes were made to the GHGs other than $CO_2$ because they were not listed [54]. Moreover, although it has been reported that $CO_2$ emissions associated with transport have decreased, the reduction in $CO_2$ emission associated with transport was not included in this study because other products were transported together with cement (Fig 3). From 2015 to 2024, the reduction in $CO_2$ emissions associated with cement production contributed to the reduction in GHG emissions associated with the concrete check dam by approximately 13–29%. In our study, the GHG emissions associated with the construction of the concrete check dam were not lower than those associated with the construction of the timber check dam. However, suppose $CO_2$ emissions associated with cement production would be significantly reduced by 2050. In that case, the results indicate that the concrete check dam may be associated with lower GHG emissions than the timber or steel check dams. As shown in Fig 4, it was indicated that GHG emissions from cement production could potentially be reduced by approximately 90% by 2050. Additionally, the overall GHG emissions from constructing concrete check dams could be reduced by around 70%. However, it should be noted that reductions in emissions due to improvements in production technology and efforts on the demand side for timber and steel were not considered in this study.

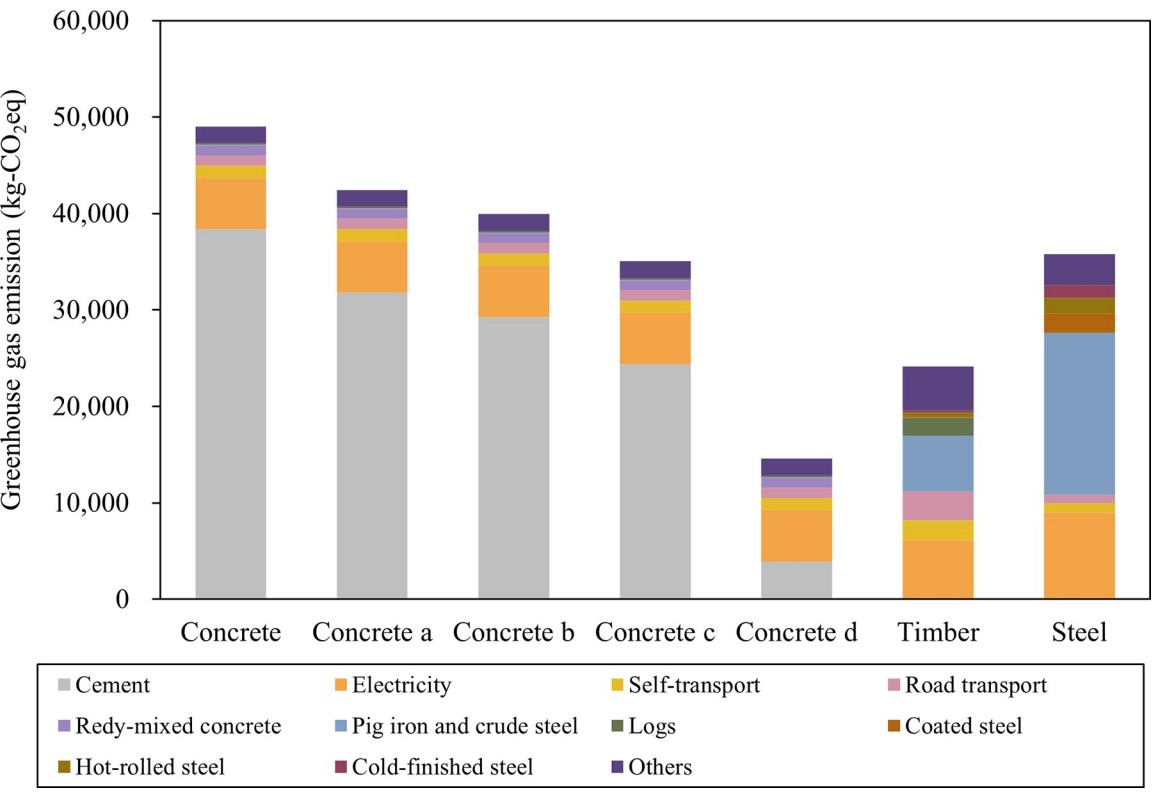

**Fig 4. Greenhouse gases and carbon dioxide emissions associated with cement production for constructing concrete check dams have decreased since 2015.**

To be sure, technically, it may be possible to reduce GHG emissions from cement production. However, while there have been reports of technical measures to reduce GHG emissions in cement production, there are few reports of policy measures [55]. As reported by Habert et al. [56], we believe that such GHG reductions cannot be achieved without the involvement of all stakeholders. While reducing GHG emissions in cement production is technically feasible, we believe specific policy discussions are needed to implement this technology.

Concrete represents greenhouse gas emissions from the concrete check dam in this study. Concrete d represents greenhouse gas emissions of the concrete check dam in 2050, as estimated using the value of the supply and demand side strategies in a previous study [54]. In this study, timber represents greenhouse gas emissions from the timber check dam. Steel represents greenhouse gas emissions from the steel check dam in this study.

## Reduction in GHG emissions

The net GHG emissions and reductions were evaluated to substantiate replacing concrete and steel with timber to construct check dams. Net reductions were expressed as negative values, while net emissions were expressed as positive values (Fig 5, S7–S9 Tables).

Concrete represents greenhouse gas emissions from the concrete check dam in this study. Concrete d represents greenhouse gas emissions of the concrete check dam in 2050, as estimated using the value of the supply and demand side strategies in a previous study [54]. In this study, timber represents greenhouse gas emissions from the timber check dam. Steel represents greenhouse gas emissions from the steel check dam in this study.

When concrete was replaced by timber as the construction material for check dams, emissions from road transportation, production of metal fittings (pig iron, crude steel) and timber (timber, logs), and electricity consumption resulted in net emissions of 14,540 kg-$CO_2$eq. It should be noted that timber and logs were not used in the concrete check dam. However, the net reduction was −39,433 kg-$CO_2$eq due to the elimination of cement and ready-mixed concrete production. Therefore, replacing a concrete check dam with a timber check dam is expected to reduce GHG emissions by 24,893 kg-$CO_2$eq.

Replacement of the steel check dam with a timber check dam resulted in net emissions of 7,540 kg-$CO_2$eq; these included emissions associated with the production of components not used in the steel check dam. However, as more steel and metal products were used to construct the steel check dam than for the timber check dam, emissions were reduced while considering the pig iron and crude steel sectors, resulting in a net reduction of 19,213 kg-$CO_2$eq. Therefore, replacing steel with timber check dams is expected to reduce GHG emissions by 11,673 kg-$CO_2$eq. The results suggest that GHG emissions can be reduced by replacing a steel check dam with a timber check dam; however, the reduction in GHG emissions when timber check dams replace steel check dams is smaller than when timber check dams replace concrete check dams.

The transition from concrete check dam d to a timber check dam in Fig 5 represents a scenario where a concrete check dam, with GHG emission reduction measures applied, is replaced by a timber check dam by 2050. The net reduction was −4,980 kg-$CO_2$eq (the net emissions were 14,540 kg-$CO_2$eq) because the significant reduction in $CO_2$ emissions associated with cement production was less than the GHG emissions associated with the production of metal fittings used for the construction of the timber check dam. The results indicate that if $CO_2$ emissions associated with cement production were significantly reduced, replacing a concrete check dam with a timber check dam would result in GHG emissions of 9,560 kg-$CO_2$eq. If $CO_2$ emissions associated with cement production decrease significantly by 2050, and if the technology for timber production does not improve, the replacement of concrete check dams with timber check dams would not result in reduced GHG emissions, indicating that in such a scenario, a concrete check dam might be better than timber check dams at mitigating GHG emissions.

## Comparison with a previous study

In a previous study by Noda et al. [28], $CO_2$, $CH_4$, and $N_2O$ emissions associated with producing materials used for constructing the timber, concrete, and steel check dams were accumulated to assess the total GHG emissions (process flow diagram). The evaluation method, target GHGs, and year of dam construction in the study by Noda et al. [28] differed from those in the present study; however, the types and sizes of check dams they analyzed in their study were similar to those in our study (Table 5).

The timber check dam analyzed by Noda et al. [28] was approximately 20% larger than that in our study, whereas the concrete and steel check dams were approximately 14% smaller than those in our study. In this study, GHG emissions were evaluated based on the production of raw materials up to the construction itself. Therefore, the evaluation processes used in the previous and the present studies were unified by excluding the emissions associated with dam maintenance, disposal, and recycling. Additionally, GHG emissions were evaluated from the production of raw materials phase to the actual construction. After unifying the scope of the evaluation process, the GHG emissions associated with the timber, concrete, and steel check dams in the previous study [28] were 20,031 kg-$CO_2$eq, 46,100 kg-$CO_2$eq, and 29,664 kg-$CO_2$eq, respectively. The check dams in the previous study [28] emitted fewer GHGs than the

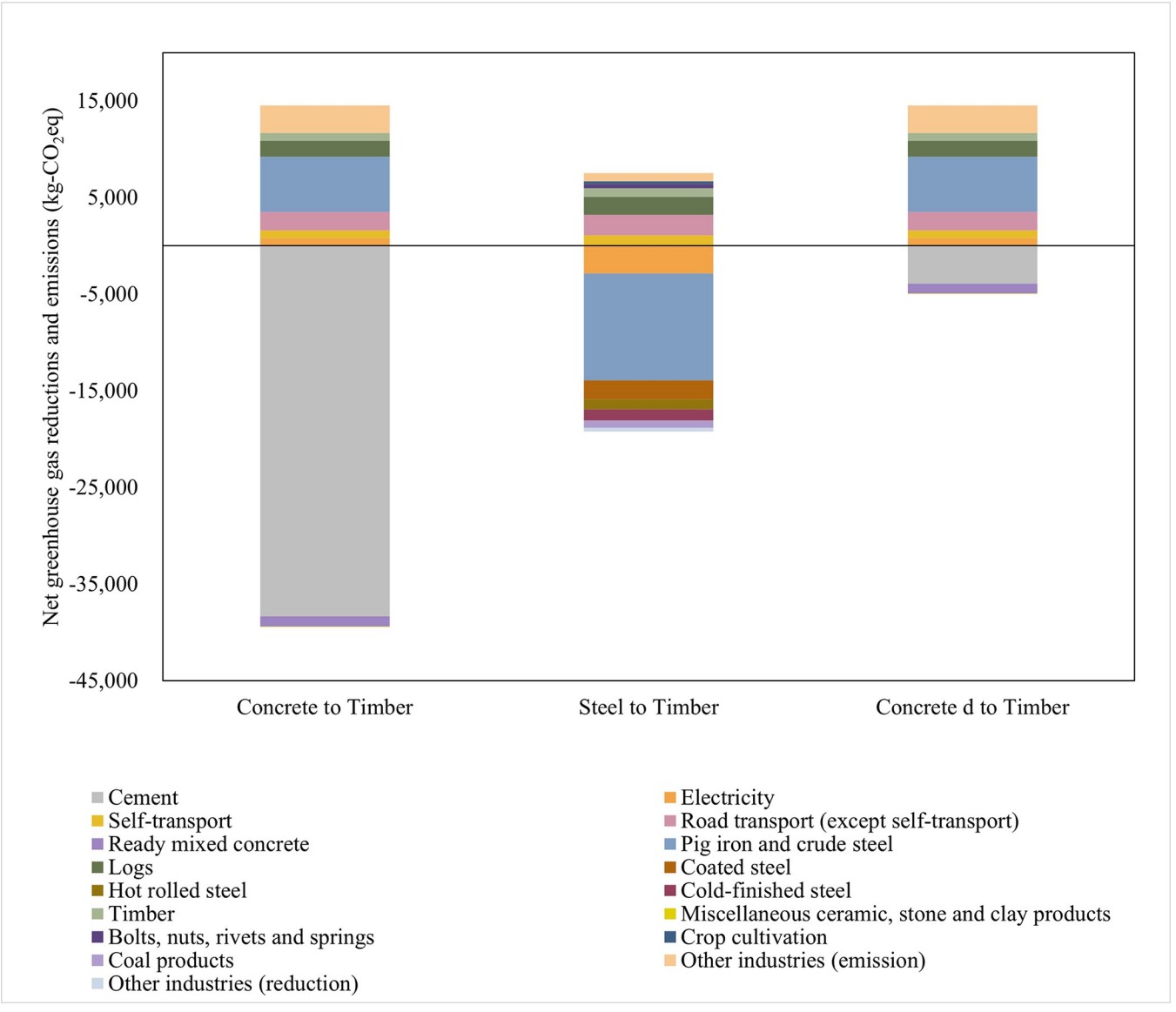

**Fig 5. Net greenhouse gas reductions and emissions associated with each industrial sector when concrete, steel, and concrete made with cement with reduced greenhouse gas emissions are replaced by timber as the primary construction material for check dams.**

**Table 5. Comparison with previous studies.**

| | This study | | Previous study [28] |
|---|---|---|---|
| Evaluation method | Input-Output analysis | | Process flow diagram analysis |
| Construction year | | 2022 | Unknown |
| Target GHG | $CO_2$, $CH_4$, $N_2O$, Hydrofluorocarbons, Organofluorine compounds, Sulfur hexafluoride, Nitrogen trifluoride | | $CO_2$, $CH_4$, $N_2O$ |
| Timber check dam scale (m$^3$) | | 156.6 | 186.9 |
| Concrete check dam scale (m$^3$) | | 150.5 | 129.4 |
| Steel check dam scale (m$^3$) | | 154.7 | 132.9 |

check dams in the present study; specifically, the previous study's timber, concrete, and steel check dams emitted approximately 17%, 6%, and 17% fewer GHGs, respectively. Although the timber check dam in the present study was smaller than that in the previous study [28], its GHG emissions were higher. As the check dams analyzed in this study were smaller than those in the previous study [28], the GHG emissions associated with the concrete and steel check dams were lower than those in the previous study [28]. The difference in GHG emissions between the timber check dam in the present study and those in the previous study [28] is attributed to the metal products (such as bolts and nuts) used. Only lag screws and deformed bars were used in the previous study; however, bolts and nuts were also used in the present study. Therefore, as more steel and metal products were used for the construction of the timber check dam in this study (compared with that in the previous study), it is expected that the GHG emissions associated with timber check dam construction will increase accordingly, and the GHG emission advantage of timber check dams may be offset.

Previous studies have also compared the environmental impact assessment using input-output and process flow diagram analyses [57–59]. In those previous studies, it has been pointed out that the results are affected by differences in evaluation methods due to differences in the number of data, system boundaries, and other factors. However, it has been evaluated that the trends of the results do not differ significantly [57]. This study also agrees with the previous studies in that the trends of the results of the two methods do not differ significantly. Compared to the process flow diagram analysis, the evaluation method using the input-output table has several advantages, including that it does not require an expensive database and that the evaluation can be performed with data from design documents. Therefore, the results of this study, which evaluated GHG emissions using the input-output table, are expected to be applied to on-site level evaluations. Thus, the results are considered to have a high return to society.

## Limitations of the work

Due to the recent reduction in GHG emissions associated with cement and ready-mixed concrete, the difference between the GHG emissions associated with constructing timber and concrete check dams may not be significant. However, even if GHG emission reductions are technically possible, whether the technology is reflected in policy is not included in the results of this study. It should be noted that this study did not verify the effect of the policy on GHG emission reductions.

## Conclusion

This study used an input-output table to evaluate GHG emissions based on the timber, concrete, and steel check dam design documents. In addition to the findings of previous studies, the superiority of timber check dams in terms of GHG emissions was evaluated. In summary, the GHG emissions associated with the timber, concrete, and steel check dams at the design stage were 24,128 kg-$CO_2$eq, 49,022 kg-$CO_2$eq, and 35,801 kg-$CO_2$eq, respectively. It was demonstrated that replacing concrete and steel check dams with a timber check dam could reduce GHG emissions by 51% and 33%, respectively. Moreover, many GHG emissions from the timber check dam could be attributed to steel production and other metal products, such as metal fittings, bolts, and nuts. The production of timber was associated with lower GHG emissions than the production of concrete and steel. However, concrete check dams may be associated with lower GHG emissions than timber check dams if GHG emissions from cement production were significantly reduced, as shown in a previous study.

The findings of this study enable the quantitative evaluation of the added value of a construction material other than its construction cost. Therefore, this study will help in the selection of timber, which has not been selected before due to its high cost, as a construction material for public structures; moreover, this study contributes to the promotion of the use of timber for construction projects to mitigate the impact of global warming.

In addition to checking dams, other structures in the civil engineering field include bridges, roadside noise barriers, and guardrails. The methodology used in this study can be used to evaluate GHG emissions from other structural design documents. Therefore, we believe evaluating the advantages of GHG emissions associated with replacing concrete and steel with timber in the civil engineering sector would be possible. Moreover, using an input-output table enables the simultaneous evaluation of regional economic impacts and GHG emissions. It is anticipated that the assessment of regional economic impacts will evolve to encompass studies that specifically analyze the economic value added by construction materials concerning their environmental impact.

## Supporting information

**S1 Table. Timber check dam: The materials and the corresponding industrial sectors.**
(PDF)

**S2 Table. Concrete check dam: The materials and the corresponding industrial sectors.**
(PDF)

**S3 Table. Steel check dam: The materials and the corresponding industrial sectors.**
(PDF)

**S4 Table. Top five sectors that contributed to the direct and indirect effects (greenhouse gas emissions in kg-$CO_2$eq) of timber check dam construction.**
(PDF)

**S5 Table. Top five sectors contributing to the direct and indirect effects (greenhouse gas emissions in kg-$CO_2$eq) of concrete check dam construction.**
(PDF)

**S6 Table. Top five sectors that contributed to the direct and indirect effects (greenhouse gas emissions in kg-$CO_2$eq) of steel check dam construction.**
(PDF)

**S7 Table. Net greenhouse gas reductions and emissions associated with each industrial sector when concrete was replaced by timber as the main construction material for check dam construction.**
(PDF)

**S8 Table. Net greenhouse gas reductions and emissions from each industrial sector when steel was replaced by timber as the main construction material for check dam construction.**
(PDF)

**S9 Table. Net greenhouse gas reductions and emissions associated with each industrial sector if the concrete check dam was replaced by a timber check dam, considering that the $CO_2$ emission associated with cement production has decreased [54].**
(PDF)

## Acknowledgments

The authors would like to thank the Akita Prefectural Department of Agriculture, Forestry and Fisheries and the Akita Forestry Consultant for providing the data analyzed in this study. We would like to express our deepest gratitude to the journal reviewers.

## Author Contributions

**Conceptualization:** Tomohumi Huzita, Ryu Noda.

**Data curation:** Tomohumi Huzita.

**Formal analysis:** Tomohumi Huzita.

**Funding acquisition:** Tomohumi Huzita.

**Investigation:** Tomohumi Huzita, Ryu Noda.

**Methodology:** Tomohumi Huzita, Ryu Noda.

**Project administration:** Tomohumi Huzita, Ryu Noda.

**Validation:** Ryu Noda.

**Visualization:** Tomohumi Huzita.

**Writing – original draft:** Tomohumi Huzita.

**Writing – review & editing:** Tomohumi Huzita, Ryu Noda.

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
