## [Decision Letter · Decision Letter 0]

19 Aug 2024

PONE-D-24-17173Comparison of greenhouse gas emissions associated with the construction of timber, concrete, and steel check dams in Akita, Japan, using input–output analysisPLOS ONE

Dear Dr. Huzita,

Thank you for submitting your manuscript to PLOS ONE. After careful consideration, we feel that it has merit but does not fully meet PLOS ONE’s publication criteria as it currently stands. Therefore, we invite you to submit a revised version of the manuscript that addresses the points raised during the review process.

We look forward to receiving your revised manuscript.

Kind regards,

Ashfaque Ahmed Chowdhury, Ph.D., FHEA, FIEB

Academic Editor

PLOS ONE

Journal Requirements: 

2. We note that [Figure 1] in your submission contain [map/satellite] images which may be copyrighted. All PLOS content is published under the Creative Commons Attribution License (CC BY 4.0), which means that the manuscript, images, and Supporting Information files will be freely available online, and any third party is permitted to access, download, copy, distribute, and use these materials in any way, even commercially, with proper attribution. For these reasons, we cannot publish previously copyrighted maps or satellite images created using proprietary data, such as Google software (Google Maps, Street View, and Earth). For more information, see our copyright guidelines: http://journals.plos.org/plosone/s/licenses-and-copyright.

Reviewers' comments:

Reviewer's Responses to Questions

**Comments to the Author**

1. Is the manuscript technically sound, and do the data support the conclusions?

Reviewer #1: Partly

Reviewer #2: Partly

2. Has the statistical analysis been performed appropriately and rigorously? 

Reviewer #1: Yes

Reviewer #2: Yes

3. Have the authors made all data underlying the findings in their manuscript fully available?

Reviewer #1: Yes

Reviewer #2: No

4. Is the manuscript presented in an intelligible fashion and written in standard English?

Reviewer #1: No

Reviewer #2: Yes

5. Review Comments to the Author

Reviewer #1: This review pertains to the article titled “Comparison of greenhouse gas emissions associated with the construction of timber, concrete, and steel check dams in Akita, Japan, using input–output analysis”.

The article is technically acceptable; however, it requires significant revision due to its writing and presentation.

Abstract:

The abstract states, “However, engineers and contractors are reluctant to select timber as a construction material because of its high construction cost”. However, this is not the sole reason for the limited use of Mass Timber in check dams. Other factors include suitability for only a limited drainage area, potential to harm grass linings in channels, reduction of hydraulic capacity, creation of turbulence which can erode channel banks, and clogging issues, particularly with leaves in the fall.

The abstract states, “However, due to the recent reduction in GHG emissions associated with cement and ready-mixed concrete, the difference between the GHG emissions associated with the construction of timber and concrete check dams may not be significant”. This point would be more appropriately addressed within the limitations section of the work, rather than in the abstract.

Introduction:

Please check the first statement in the introduction “The Intergovernmental Panel on Climate Change’s (IPCC’s) Special Report on Global Warming of 1.5 °C, which was released in October 2018, recommended that net greenhouse gas (GHG) emissions must be reduced to zero by the early 2050s or early 2070s to limit global warming to 1.5 °C or 2 °C”.

I think it is proposed that the goal should be to limit global warming to 1.5°C or 2°C by the end of 2050 and achieve net zero emissions by 2070.

Results and Discussion

The article's format and writing quality are inadequate, giving the impression that analyzed data has been indiscriminately inserted into the text. To enhance comprehension and visual appeal, please incorporate diagrams and present data in supplementary materials.

Comparisons with previously published work should be approached with caution, check if prior studies have employed similar assumptions or different.

The utility of this method is questionable when more precise techniques, like process flow diagram analysis, are accessible.

Reviewer #2: It is an excellent topic, comparing structures from an environmental point of view, increases knowledge and allows other researchers to understand the benefits of materials in applications.

To improve the methodology, it is necessary to include a flow chart of the inputs and outputs of the three alternatives. (Example Fig. 1 https://doi.org/10.3390/buildings12030311)

It is necessary to add a contribution analysis in each alternative, I recommend improving Fig. 3, adding this contribution analysis (Example Fig. 14a https://doi.org/10.1016/j.heliyon.2024.e25056).

Introduction and discussion need to be improved, seeking more references for individual materials (cement, concrete, steel, wood). Here are some references:

https://doi.org/10.1016/j.buildenv.2014.04.025

https://doi.org/10.1007/978-94-007-7811-5_8

https://doi.org/10.1016/j.jclepro.2015.11.078

https://doi.org/10.3390/buildings12030311

https://doi.org/10.1016/j.heliyon.2024.e25056

https://doi.org/10.1016/j.buildenv.2021.108285

6. PLOS authors have the option to publish the peer review history of their article (what does this mean?). If published, this will include your full peer review and any attached files.

Reviewer #1: No

Reviewer #2: No

---

## [Author Response · Author response to Decision Letter 0]

21 Oct 2024

Response to Reviewer Comments

We appreciate the reviewers’ valuable comments on our paper, “Comparison of greenhouse gas emissions associated with the construction of timber, concrete, and steel check dams in Akita, Japan, using input-output analysis.”

We have verified that the PLOS ONE's style requirements have been met.

2. We note that [Figure 1] in your submission contain [map/satellite] images which may be copyrighted.

There is no problem because the map is from the Geospatial Information Authority of Japan (GSI). However, in accordance with PlosOne's policy, the original Fig. 1 has been removed.

3. Review Comments to the Author

This review pertains to the article titled “Comparison of greenhouse gas emissions associated with the construction of timber, concrete, and steel check dams in Akita, Japan, using input–output analysis”.

The article is technically acceptable; however, it requires significant revision due to its writing and presentation.

Comment 1

Abstract:

The abstract states, “However, engineers and contractors are reluctant to select timber as a construction material because of its high construction cost”. However, this is not the sole reason for the limited use of Mass Timber in check dams. Other factors include suitability for only a limited drainage area, potential to harm grass linings in channels, reduction of hydraulic capacity, creation of turbulence which can erode channel banks, and clogging issues, particularly with leaves in the fall.

Response 1

You have pointed out a major reason why Mass Timber cannot be used for a check dam. In this revision, we have considered your comment and included what the Japanese Forestry Agency considers as a condition for installation (Manuscript.docx line 21 -23).

Comment 2

The abstract states, “However, due to the recent reduction in GHG emissions associated with cement and ready-mixed concrete, the difference between the GHG emissions associated with the construction of timber and concrete check dams may not be significant”. This point would be more appropriately addressed within the limitations section of the work, rather than in the abstract.

Response 2

The information has been removed from the abstract. In addition, a subsection on the limitations of the study has been prepared and added after the discussion section (Manuscript.docx line 459 - 465).

Comment 3

Introduction:

Please check the first statement in the introduction “The Intergovernmental Panel on Climate Change’s (IPCC’s) Special Report on Global Warming of 1.5 °C, which was released in October 2018, recommended that net greenhouse gas (GHG) emissions must be reduced to zero by the early 2050s or early 2070s to limit global warming to 1.5 °C or 2 °C”.

I think it is proposed that the goal should be to limit global warming to 1.5°C or 2°C by the end of 2050 and achieve net zero emissions by 2070.

Response 3

As you have pointed out, the wording was inappropriate. We have double-checked and corrected the content (Manuscript.docx line 38 - 42).

Comment 4

Results and Discussion

The article's format and writing quality are inadequate, giving the impression that analyzed data has been indiscriminately inserted into the text. To enhance comprehension and visual appeal, please incorporate diagrams and present data in supplementary materials.

Response 4

The discussion has been improved by citing new references (References 5, 6, 7, 12, 13, 14, 15, 16, 17, 48, 49, 52, 53, 55, 56, 57, 58, 59). Furthermore, we have visually presented the results using Figs. 3-5 and added supplementary material (Tables S1-S9).

Comment 5

Comparisons with previously published work should be approached with caution, check if prior studies have employed similar assumptions or different.

Response 5

In comparison with previous studies, we have again identified similarities and differences between the present and previous studies. We have summarized our findings in Table 5 so that they can be conveyed clearly to the reader.

Comment 6

The utility of this method is questionable when more precise techniques, like process flow diagram analysis, are accessible.

Response 6

We believe that the strengths of this study lie in the fact that the methods are easy to execute for local governments and businesses, including the fact that it does not use an expensive database, making GHG emissions data more accessible to the public. The content has been revised with new wording (Manuscript.docx line 446-457).

4. Review Comments to the Author

It is an excellent topic, comparing structures from an environmental point of view, increases knowledge and allows other researchers to understand the benefits of materials in applications.

Comment 1

To improve the methodology, it is necessary to include a flow chart of the inputs and outputs of the three alternatives. (Example Fig. 1 https://doi.org/10.3390/buildings12030311)

Response 1

A flow chart has been created for the three dams based on the references provided (Fig. 1).

Comment 2

It is necessary to add a contribution analysis in each alternative, I recommend improving Fig. 3, adding this contribution analysis (Example Fig. 14a https://doi.org/10.1016/j.heliyon.2024.e25056).

Response 2

With the help of the references we received, we have modified the Original Fig. 3 to show the contribution analysis (Fig. 4). We have also modified the discussion to show each industry's contribution (Manuscript.docx line 341-344).

Comment 3

Introduction and discussion need to be improved, seeking more references for individual materials (cement, concrete, steel, wood). Here are some references:

https://doi.org/10.1016/j.buildenv.2014.04.025

https://doi.org/10.1007/978-94-007-7811-5_8

https://doi.org/10.1016/j.jclepro.2015.11.078

https://doi.org/10.3390/buildings12030311

https://doi.org/10.1016/j.heliyon.2024.e25056

https://doi.org/10.1016/j.buildenv.2021.108285

Response 3

We have added references for materials and discussion (References 5, 6, 7, 12, 13, 14, 15, 16, 17, 48, 49, 52, 53, 55, 56, 57, 58, 59) and improved the introduction and discussion.

Thank you again for your insightful comments on our paper. We hope we have sufficiently addressed your comments and we are eager to move this revised manuscript closer to publication.

---

## [Decision Letter · Decision Letter 1]

8 Dec 2024

Comparison of greenhouse gas emissions associated with the construction of timber, concrete, and steel check dams in Akita, Japan, using input–output analysis

PONE-D-24-17173R1

Dear Dr. Huzita,

We’re pleased to inform you that your manuscript has been judged scientifically suitable for publication and will be formally accepted for publication once it meets all outstanding technical requirements.

Kind regards,

Ashfaque Ahmed Chowdhury, Ph.D., FHEA, FIEB

Academic Editor

PLOS ONE

Additional Editor Comments:

The title of the paper can be slightly amended as follows: 

Comparison of greenhouse gas emissions associated with the construction of timber, concrete, and steel check dams in Akita, Japan: an input–output analysis

Reviewers' comments:

Reviewer's Responses to Questions

**Comments to the Author**

1. If the authors have adequately addressed your comments raised in a previous round of review and you feel that this manuscript is now acceptable for publication, you may indicate that here to bypass the “Comments to the Author” section, enter your conflict of interest statement in the “Confidential to Editor” section, and submit your "Accept" recommendation.

Reviewer #1: All comments have been addressed

Reviewer #2: All comments have been addressed

2. Is the manuscript technically sound, and do the data support the conclusions?

Reviewer #1: Yes

Reviewer #2: Yes

3. Has the statistical analysis been performed appropriately and rigorously? 

Reviewer #1: Yes

Reviewer #2: Yes

4. Have the authors made all data underlying the findings in their manuscript fully available?

Reviewer #1: Yes

Reviewer #2: Yes

5. Is the manuscript presented in an intelligible fashion and written in standard English?

Reviewer #1: No

Reviewer #2: Yes

6. Review Comments to the Author

Reviewer #1: The authors have updated the manuscript based on the recommendations. The manuscript can be accepted for publication.

Reviewer #2: The manuscript has been improved according to the reviewers comments. With these new adds, readers could understand better the LCA scope and results.

7. PLOS authors have the option to publish the peer review history of their article (what does this mean?). If published, this will include your full peer review and any attached files.

Reviewer #1: **Yes: **Vaibhav Kumar

Reviewer #2: No

---

## [Editor Report · Acceptance letter]

3 Jan 2025

PONE-D-24-17173R1 

PLOS ONE

Dear Dr. Huzita, 

I'm pleased to inform you that your manuscript has been deemed suitable for publication in PLOS ONE. Congratulations! Your manuscript is now being handed over to our production team.

Kind regards, 

on behalf of

Dr. Ashfaque Ahmed Chowdhury 

Academic Editor

PLOS ONE